# Survivin Splice Variants in Arsenic Trioxide (As_2_O_3_)-Induced Deactivation of PI3K and MAPK Cell Signalling Pathways in MCF-7 Cells

**DOI:** 10.3390/genes10010041

**Published:** 2019-01-14

**Authors:** Kagiso Laka, Lilian Makgoo, Zukile Mbita

**Affiliations:** Department of Biochemistry, Microbiology and Biotechnology, University of Limpopo, Private Bag X1106, Sovenga 0727, Polokwane, South Africa; lakakagiso@gmail.com (K.L.); lilianmakgoo@gmail.com (L.M.)

**Keywords:** breast cancer, survivin splice variants, arsenic trioxide, cell cycle, apoptosis

## Abstract

Several pathways are deregulated during carcinogenesis but most notably, tumour cells can lose cell cycle control and acquire resistance to apoptosis by expressing a number of anti-apoptotic proteins such as the Inhibitors of Apoptosis Protein (IAP) family of proteins that include survivin, which is implicated in cancer development. There is no study which had proven that arsenic trioxide (As_2_O_3_) has any effect on the splicing machinery of survivin and its splice variants, hence this study was aimed at determining the cytotoxic effect of As_2_O_3_ and its effect on the expression pattern of survivin splice variants in MCF-7 cells. As_2_O_3_ inhibited the growth of the MCF-7 cells in a concentration-dependent manner. The Muse^®^ Cell Analyser showed that As_2_O_3_-induced G2/M cell cycle arrest, promoted caspase-dependent apoptosis without causing any damage to the mitochondrial membrane of MCF-7 cells. As_2_O_3_ also deactivated two survival pathways, Mitogen-Activated Protein Kinase (MAPK) and Phosphoinositide 3-Kinase (PI3K) signalling pathways in MCF-7 cells. Deactivation of the two pathways was accompanied by the upregulation of survivin 3α during As_2_O_3_-induced G2/M cell cycle arrest and apoptosis. Survivin 2B was found to be upregulated only during As_2_O_3_-induced G2/M cell cycle arrest but downregulated during As_2_O_3_-induced apoptosis. Survivin wild-type was highly expressed in the untreated MCF-7 cells, the expression was upregulated during As_2_O_3_-induced G2/M cell cycle arrest and it was downregulated during As_2_O_3_-induced apoptosis. Survivin variant ΔEx3 was undetected in both untreated and treated MCF-7 cells. Survivin proteins were localised in both the nucleus and cytoplasm in MCF-7 cells and highly upregulated during the As_2_O_3_-induced G2/M cell cycle arrest, which can be attributed to the upregulation of survivin-2B. This study has provided the first evidence showing that the novel survivin 2B splice variant may be involved in the regulation of As_2_O_3_-induced G2/M cell cycle arrest only. This splice variant can therefore, be targeted for therapeutic purposes against Luminal A breast cancer cells.

## 1. Introduction

According to the National Cancer Registry (NCR), more than 100,000 South Africans are diagnosed with cancer each year, of which 21% is attributed to breast cancer [1]. Development of breast cancer is characterized by the deregulation of cell cycle control and resistance to apoptosis [2]. Tumour cells can acquire resistance to apoptosis by expressing a number of anti-apoptotic proteins such as the Inhibitors of Apoptosis Protein (IAP) family of proteins that include survivin [3]. These proteins inhibit apoptosis by hindering caspase cascade and therefore, inhibiting both intrinsic and extrinsic pathways [4]. The resistance to apoptosis is multifactorial and there are several role players that can be implicated in caspase inhibition or independent. As reviewed in Mohammad et al. [5], Bcl-2 family of proteins, autophagy, proteasome pathway and epigenetics all play a role in resistance to apoptosis.

Although several efforts have been made to reduce the death rates associated with cancer, to date, there has been no ultimate chemotherapeutic drug available to successfully overcome this disease. Targeting biomolecules that regulate molecular mechanisms has become popular and survivin has attracted a lot of research interest across the globe due to the discovery of a wide range of splice variants with diverse biological functions [4,6]. Survivin, encoded by the *Baculoviral IAP repeat containing 5 (BIRC5*) gene, is the smallest and best studied member of the IAPs family, which is involved in the regulation of cell division, promoting angiogenesis, inhibition of both caspase-dependent and caspase-independent apoptosis in cancer cells [7,8]. Survivin wild-type is strongly expressed during embryonic and foetal development but rarely found in normal adult tissues [4]. It is, however, up-regulated in transformed cancer cell lines and most common types of human cancers [9,10], where it promotes tumour survival by reducing apoptosis as well as favouring endothelial cell proliferation and migration.

Alternative splicing of the *BIRC5* gene produces six survivin splice variants, namely, wild type survivin, survivin 2B, survivin 2α, survivin 3B, survivin ∆Ex3 and survivin 3α [6]. Survivin has been acknowledged as an essential molecular marker and target in a range of cancer diagnosis and therapeutics [11].

As_2_O_3_ has been shown to exert anticancer activities against solid cancers, including breast cancer [12,13]. As_2_O_3_ has also been demonstrated to inhibit lung adenocarcinoma cell line (H1355) growth by down-regulating survivin expression and through the activation of p38 and c-Jun N-terminal kinases (JNK) pathways [14]. Inhibition of Phosphoinositide 3-Kinase (PI3K) or extracellular signal-regulated kinases (ERK) signalling led to clear inhibition of survivin expression. However, pre-treatment with p38 Mitogen-Activated Protein Kinase (MAPK) inhibitor led to up-regulated survivin levels.

The role and the expression of the survivin splice variants are not fully understood and there is no study which had proven that As_2_O_3_ has any effect on the splicing machinery of survivin and its splice variants. This study focused on analysing the expression pattern of the different survivin splice variants during both As_2_O_3_-induced apoptosis and cell cycle arrest in breast cancer MCF-7 cells.

## 2. Methods and Materials

### 2.1. Materials

The MCF-7 cells were kindly donated by Prof Mervin Meyer from the University of the Western Cape, South Africa. Dulbecco’s Modified Eagle Medium (DMEM) and foetal bovine serum (FBS) were purchased from Hyclone (Hyclone, South Logan, UT, USA). Antibiotic mixture containing penicillin and streptomycin (Pen-Strep), phosphate buffered saline (PBS) the MTT [3-(4,5-dimethylthiazol-2-yl)-2, 5-diphenyltetrazolium bromide], 4′, 6-diamidino-2-phenylindole (DAPI), Trizol reagent were obtained from ThermoFisher Scientific (ThermoFischer Scientific, Waltham, MA, USA) while Dimethyl Sulfoxide (DMSO) was purchased from (Sigma-Aldrich (Sigma-Aldrich, St. Louis, MO, USA). The AMV II Reverse transcription system was purchased from Promega (Promega, Madison, WI, USA) while the EmeraldAmp^®^ GT-PCR Kit was procured from Takara Bio (Takara Bio, Kasatsu, Shiga Prefecture, Japan). The Muse^®^ Assay Kits (Muse^®^ Count and Viability Assay, Muse^®^ Cell Cycle Assay, Muse^®^ Annexin V and Dead Cell Assay, Muse^®^ MitoPotential Assay, Muse^®^ Multi-Caspase Assay, Muse^®^ MAPK Assay and Muse^®^ PI3k Assay) were all purchased from Merck-Millipore (Merck-Millipore, Darmstadt, Germany). All the reagents were used without further purification or alterations.

### 2.2. Cell Culture

MCF-7 cells were cultured in DMEM supplemented with 10% FBS and 1% antibiotic mixture of Penicillin and Streptomycin and maintained in culture flasks at 37 °C in a humidified chamber containing 5% CO_2_.

### 2.3. Cytotoxicity Assay

The MCF-7 cells viability was tested by the MTT assay to evaluate the cytotoxicity of the As_2_O_3_. Briefly, the MCF-7 cells were diluted into a single cell suspension and 2 × 10^3^ cells/well were seeded in 96-well culture plates and allowed to attach, overnight. The cells were washed with 1 × PBS, then treated with different concentrations of As_2_O_3_, cobalt chloride and curcumin for 24 h. After 24 h, the treatment was discarded and wells were washed with 1 × PBS. Then, 10 µL of MTT reagent (5 mg/mL) was added to each well and the plates were incubated for 4 h in the CO_2_ incubator. Following incubation, 100 µL DMSO was added to dissolve formazan crystals and the absorbance readings were taken at 560 nm using a microplate reader (Promega). The cell viability was assessed using the formula:$$\mathrm{Cell}\;\mathrm{viability}\;(\%)=\frac{\mathrm{Average}\;\mathrm{OD}\;(\mathrm{experimental}\;\mathrm{group})}{\mathrm{Average}\;\mathrm{OD}\;(\mathrm{untreated}\;\mathrm{group})}\times 100\%$$

The IC_50_s were extrapolated from the graph of cell viability versus the concentration of the compounds and confirmed using the MUSE^®^ Cell Viability Assay. Cobalt chloride was used as a positive control for cell cycle assays only as it is known to induce the G2/M cell cycle arrest while curcumin was used as a positive control for apoptosis. These two positive controls were therefore used for the relevant experiments.

### 2.4. Muse^®^ Count and Viability Assay

In order to confirm the cell cycle arrest and apoptotic inducing concentrations, MCF-7 cells (2 × 10^5^ cells/well) were cultured in six well plates and treated with 11 and 32 µM As_2_O_3_, 100 µM cobalt chloride and 100 µM curcumin for 24 h. Following, treatment, the cells were trypsinized and centrifuged at 300× *g* for 5 min. The cells were then resuspended in 20 µL cell culture medium and 380 µL Muse^®^ Count and Viability Reagent (Merck-Millipore) was added to each sample. The samples were incubated for 5 min in the dark at room temperature and then analysed using the Muse^®^ Cell Analyser (Merck-Millipore).

### 2.5. Light and Fluorescence Microscopic Analysis

MCF-7 cells (5 × 10^4^ cells/well) were seeded in 24 well plates, overnight. After incubation, the cells were then exposed to various concentrations of As_2_O_3_ (0, 4, 11, 16 and 32 µM), cobalt chloride (50 and 100 µM) and curcumin (50 and 100 µM) for 24 h. After treatment, the cells were washed with sterile 1 × PBS and fixed with 3.7% paraformaldehyde for 10 min at room temperature. The cells were then washed twice with sterile 1 × PBS and stained with DAPI (5 µg/mL) for 15 min in the dark, at room temperature. Following incubation, the cells were washed twice with sterile 1 × PBS. Morphological changes of MCF-7 cells in each group were observed under the Olympus CKX53 Inverted Microscope, (Olympus, Shinjuku, Tokyo, Japan) and Eclipse Ti-U fluorescence microscope (Nikon Instruments Inc., Melville, NY, USA), for light and fluorescence microscopy, respectively. The images were captured with DSRI-1 camera and an LC-30 camera set, respectively.

### 2.6. Cell Cycle Assay

To confirm whether As_2_O_3_ induces cell cycle arrest in MCF-7 cells, Muse^®^ Cell Cycle Kit was used. MCF-7 cells (2 × 10^5^ cells/well) were cultured in six well plates and allowed to settle, overnight. The cells were then treated with 11 µM of As_2_O_3_ and 100 µM of cobalt chloride and incubated for 24 h in appropriate cell culture conditions mentioned above. Following incubation, the cells were mildly trypsinized and centrifuged at 300× *g* for 5 min. The cells were then washed with sterile 1 × PBS and fixed for 3 h in 70% ethanol at −20 °C. Following incubation, the cells were centrifuged, washed with sterile 1 × PBS and 200 µL of the Muse^®^ Cell Cycle reagent was added and incubated in the dark for 30 min. The samples were analysed using the Muse^®^ Cell Analyser (Merck-Millipore).

### 2.7. Annexin V and Dead Cell Assay

To confirm whether As_2_O_3_ induces apoptosis in MCF-7 cells, the Annexin V and Dead Cell Kit (MerkMillipore, Darmstadt, Germany) was used. MCF-7 Cells (2 × 10^5^ cells/well) were seeded and allowed to adhere in six well plates, overnight. The cells were treated with 32 µM of As_2_O_3_ and 100 µM curcumin, which was used as a positive control, for 24 h. After the incubation period, the cells were trypsinized and centrifuged at 300× *g* for 5 min. The cells were re-suspended in 100 µL of DMEM with 1% FBS and 100 µL of Annexin V and Dead Cell Reagent was added to each tube. Each sample was vortexed for 5 s and incubated in the dark at room temperature for 20 min. Samples were then analysed using the Muse^®^ Cell Analyser (Merck-Millipore).

### 2.8. Mitochondrial Membrane Potential Assay

Mitochondrial membrane depolarization, a marker of apoptosis, was detected using the Muse^®^ MitoPotential Assay kit according to the manufacturer’s protocol (Merck-Millipore). Briefly, 2 × 10^5^ cells/well were cultured in six well plates and allowed to adhere to the bottom of the plate, overnight. Following incubation, the cells were treated with 32 µM As_2_O_3_ and 100 µM curcumin. After 24 h treatment, the cells were trypsinized and centrifuged at 300× *g* for 5 min. The cell suspension for each sample was mixed with 95 µL MitoPotential dye and incubated for 20 min at 37 °C. After incubation, 5 µL MitoPotential 7-AAD was and placed in the dark for 5 min. The samples were then analysed using the Muse^®^ Cell Analyzer (Merck-Millipore).

### 2.9. Multi-Caspase Assay

To classify apoptosis induced by As_2_O_3_, the Multi-Caspase Assay was performed as instructed by the Muse^®^ Multi-Caspase Kit protocol (Merck Millipore). MFC-7 cells (2 × 10^5^ cells/well) were cultured in six well plates and allowed to adhere to the bottom of the plate, overnight. Following the incubation overnight, the cells were treated with 32 µM As_2_O_3_ and 100 µM curcumin for 24 h, in appropriate cell culture conditions. Following treatment, the cells were trypsinized and centrifuged at 300× *g* for 5 min. Multi-Caspase buffer was used to re-suspend the cells and 50 µL of each sample was transferred to a new tube and 5 µL of Multi-Caspase reagent was added. Samples were mixed and incubated at 37 °C in CO_2_ incubator for 30 min, followed by addition of 150 µL 7-AAD and placed in the dark for 5 min at room temperature. The samples were then analysed using the Muse^®^ Cell Analyser (Merck-Millipore).

### 2.10. MAPK Dual Pathway Activation Assay

The Muse^®^ MAPK Assay was employed to detect the total protein expression and to measure the phosphotransferase activity of the mitogen-activated protein kinase according to the manufacturer (Merck-Millipore). The cells were seeded (1 × 10^5^ cells/well) in six well plates and allowed to adhere overnight in a 37 °C humidified incubator containing 5% CO_2_. The cells were then treated with 11 and 32 µM As_2_O_3_, 100 µM cobalt chloride and 100 µM curcumin for 24 h. Following treatment, the cells were trypsinized and centrifuged at 300× *g* for 5 min. The cells were resuspended in 500 µL of 1 × MAPK buffer and equal amounts of fixation buffer were added. The samples were placed on ice for 5 min. The samples were centrifuged at 300× *g* for 5 min, permeabilized by adding 1 mL of ice-cold permeabilization buffer and placed on ice for 5 min. Following incubation, the cells were centrifuged at 300× *g* for 5 min, resuspended in 450 µL of 1 × MAPK buffer and 10 µL of antibody working cocktail (5 µL of anti-phospho-ERK1/2 (Thr202/Thr204, Thr185/Thr187), phycoerythrin and 5 µL of anti-ERK1/2 PECy5 conjugated antibodies) and incubated for 30 min in the dark at room temperature. Following incubation step, 100 µL of 1 × MAPK buffer was added to each sample and centrifuged at 300× *g* for 5 min. The cells were resuspended in 200 µL of 1 × MAPK buffer and analysed using Muse^®^ Cell Analyser (Merck-Millipore).

### 2.11. PI3K Dual Pathway Activation Assay

The Muse^®^ phosphoinositide 3-kinase (PI3K) Assay was performed to detect the extent of Akt phosphorylation relative to the total Akt expression in given cell population according to the manufacturer (Merck-Millipore). Similar to MAPK Assay, the cells (2 × 10^5^ cells/well) were seeded in six well plates and allowed to attach, overnight in a 37 °C humidified incubator containing 5% CO_2_. The cells were then treated with 11 and 32 µM As_2_O_3_, 100 µM cobalt chloride and 100 µM curcumin for 24 h. Following treatment, the cells were trypsinized and centrifuged at 300× *g* for 5 min. The cells were resuspended in 500 µL of 1 × PI3K buffer and equal amounts of fixation buffer were added. The samples were placed on ice for 5 min. The samples were centrifuged at 300× *g* for 5 min, permeabilized by adding 1 mL of ice-cold permeabilization buffer and placed on ice for 5 min. Following incubation, the cells were centrifuged at 300× *g* for 5 min, resuspended in 450 µL of 1 × PI3K buffer and 10 µL of antibody working cocktail (5 µL of anti-phospho-Akt [Ser473], Alexa, Fluor^®^555 and 5 µL of anti-Akt/PKB, PECy5 conjugated antibodies) and incubated for 30 min in the dark at room temperature. Following incubation step, 100 µL of 1 × PI3K buffer was added to each sample and centrifuged at 300× *g* for 5 min. The cells were resuspended in 200 µL of 1 × PI3K buffer and analysed using Muse^®^ Cell Analyser (Merck-Millipore).

### 2.12. Polymerase Chain Reaction

Total RNA was isolated from treated and untreated MCF-7 cells using TRIzol reagent from ThermoFisher Scientific, USA. Complementary deoxyribonucleic acid (cDNA) was synthesized from the total RNA using the AMV II Reverse transcription System (Promega, USA). The cDNA samples were normalized to a consistent concentration of 200 ng. The cDNA samples were then subjected to PCR using primers which amplify the different survivin splice variants (Table 1).

The EmeraldAmp^®^ GT PCR Kit (Takara Bio, USA) was used and the manufacturer’s instructions were followed. Briefly, the 25 µL reaction mix containing 12.5 µL EmeraldAmp^®^, GT PCR Master Mix (Takara Bio), 1 µl of each primer, 9.5 µl of water and 1 µl of DNA template. The PCR products were visualised on a 1.3% agarose gels. The gels were then viewed using a Chemidoc XRS image analyser (BioRad, Hercules, CA, USA).

### 2.13. Immunocytochemistry

To view the localization and expression of survivin proteins after As_2_O_3_ treatment, the cells were subjected to immunofluorescence staining. The MCF-7 cells were treated with 11 µM and 32 µM As_2_O_3_, 100 µM cobalt chloride and 100 µM curcumin for 24 h at 37 °C in a humidified atmosphere. The cells were washed with sterile 1 X PBS and fixed with 4% of paraformaldehyde for 15 min. After fixing, the cells were permeabilized with 0.25% Triton™ × -100 for 10 min and antibody non-specific binding was blocked using 5% BSA for 1 h at room temperature. The cells were labelled with 5 µg/mL survivin recombinant rabbit monoclonal antibody (Product # 700387) [ThermoFischer Scientific, USA] for 1 h at room temperature. The cells were then washed with sterile 1 × PBS, then labelled with 4 µg/mL Alexa Flour^®^ 488 goat anti-rabbit IgG secondary antibody (Product # A-11008) [ThermoFischer Scientific, US] for 30 min in the dark. The cell nuclei were identified by counterstaining with DAPI for 5 min. The samples were examined using an Eclipse Ti-U fluorescence microscope (Nickon Instruments Inc.) at 20 × magnification and images were captured with DSRI-1 camera.

### 2.14. Statistical Significance

All data was analysed using Graph pad prism version 6.0 statistical software and the values are expressed as mean ± standard error of mean (SEM). Statistical significance was considered at *p* < 0.05 using the Turkey Cramer multiple comparison test. The asterisk (*) (**) and (***) show *p* < 0.05, *p* < 0.01 and *p* < 0.001, respectively.

## 3. Results

### 3.1. As_2_O_3_ Reduced the Viability of MCF-7 Cells in a Concentration Dependent Manner

The MTT Assay showed that the treatment of MCF-7 cells with different concentrations of As_2_O_3_, cobalt chloride and curcumin for 24 h significantly (*p* < 0.001) reduced the viability of MCF-7 cells in a concentration-dependent manner (Figure 1a–c). The IC_50_ for As_2_O_3_ was found to be 11 µM (51.78 ± 1.243) and was used for further studies whereas, 100 µM cobalt chloride (47.30 ± 2.610) and curcumin (50.32 ± 3.213) were used throughout the study as positive controls.

### 3.2. Confirmation of the IC_50_s Using the MUSE^®^ Count and Viability Assay

The MTT results and IC_50_s obtained were confirmed by using the Muse^®^ Count and Viability Assay. As shown in Figure 2b–e, the treatment of MCF-7 cells with As_2_O_3_ (11 µM and 32 µM), cobalt chloride (100 µM) and curcumin (100 µM) reduced the viability of MCF-7 cells (95.08 ± 0.851) to 51.78 ± 1.243 (*p* < 0.001) and 29.45 ± 2.563 (*p* < 0.001), 46.22 ± 2.408 and 28.76 ± 2.386 [Figure 2f], respectively.

### 3.3. As_2_O_3_ Induced Typical Apoptotic Morphological Changes and Growth Inhibition of the MCF-7 Cells

To confirm the effect of As_2_O_3_ on the morphology and growth of the breast cancer MCF-7 cells. The cells were treated for a period of 24 h and then analysed using the inverted light microscope (Figure 3). The untreated cells (Figure 3a) maintained their normal epithelial shape (yellow arrows). However, the cells treated with As_2_O_3_ (Figure 3b–e) exhibited morphological changes typical to apoptosis features such as cell shrinkage (red arrows). Additionally, the cell numbers were reduced as the concentrations of As_2_O_3_, cobalt chloride and curcumin increased.

### 3.4. As_2_O_3_ Induces Mitotic and Apoptotic Morphological Changes in MCF-7 Cells

To analyse the nuclear morphological changes in As_2_O_3_-treated MCF-7 cells, DAPI staining was used. As shown in Figure 4a, DAPI staining showed that the untreated cells maintained the normal MCF-7 morphology which includes intact nuclear structure. As_2_O_3_ induced cell cycle arrest in MCF-7 cells as demonstrated by accumulation of cells at anaphase (Figure 4b) indicated by white arrows in 4µM treated cells. The cells treated with 16 and 32 µM As_2_O_3_ (Figure 4d,e) and 100 µM cobalt chloride (Figure 4f) and 100 µM curcumin (Figure 4g), showed hallmark features of apoptosis such as chromatin condensation and fragmentation of the nucleus as indicated by red arrows.

### 3.5. As_2_O_3_ Induced G2/M Cell Cycle Arrest in Treated MCF-7 Cells

To confirm the cell cycle arrest induced by As_2_O_3_ as demonstrated in Figure 4b, c, we examined the DNA content using the Muse^®^ Cell Cycle Assay Kit. As_2_O_3_ (11 µM) significantly (*p* < 0.001) induced the G2/M cell cycle arrest in the MCF-7 breast cancer cells (Figure 5). Comparing with the untreated cells (25.73 ± 1.824), 11 µm of As_2_O_3_ (45.13 ± 0.942; *p* < 0.001) and 100 µM cobalt chloride (35.50 ± 1.428; *p* < 0.05) significantly increased the percentage of cells at G2M cell cycle phase.

### 3.6. As_2_O_3_ Promotes Programmed Cell Death of MCF-7 Cells

To further confirm that As_2_O_3_-induced cellular apoptosis in treated MCF-7 cells, the Annexin V and Dead Cell Kit and Muse^®^ Cell Analyser were employed (Figure 6a–e). As_2_O_3_ induced apoptosis of MCF-7 breast cancer cells after 24 h treatment. As_2_O_3_ significantly (*p* < 0.01) increased the population of apoptotic cells up to 73.43 ± 6.045 comparing to the untreated cells which showed 10.69 ± 1.451 (Figure 6f).

### 3.7. As_2_O_3_ Does Not Disrupt the Mitochondrial Membrane of MCF-7 Cells

To determine if As_2_O_3_ induces the intrinsic or extrinsic pathway, MCF-7 cells were treated with As_2_O_3_ and curcumin. Few live (1.22 ± 0.362) and dead (0.17 ± 0.078) cells had mitochondrial damaged after treatment with 32 μM As_2_O_3_. Comparing with the untreated cells (2.84 ± 0.438), As_2_O_3_ (1.45 ± 0.475) caused less damage to the mitochondrial membranes of MCF-7 cells as demonstrated in Figure 7, respectively.

### 3.8. Multi-Caspase Activation of MCF-7 Cells

To determine whether the As_2_O_3_-induced apoptosis was caspase-dependent or caspase-independent, the Multi-Caspase Assay was used. We observed that 32 µM of As_2_O_3_ significantly induced the activation of several caspases including (1, 3, 4, 5, 6, 7, 8 and 9), to 94.07 ± 4.324 (*p* < 0.001), while curcumin induced caspases activation to 87.29 ± 5.483 (*p* < 0.001) comparing with the untreated cells which had 8.62 ± 1.336 as shown in Figure 8, respectively.

### 3.9. As_2_O_3_ Downregulated MAPK Activation in MCF-7 Cells

To analyse the mechanism by which As_2_O_3_ inhibits cell growth, we investigated whether As_2_O_3_ exerts its anti-tumour activity through inactivation of MAPK and PI3K signalling pathways. The mitogen-activated protein kinase (MAPK) pathway is a signalling cascade activated by pro-inflammatory stimuli and cellular stresses, playing a critical role in the translational regulation of pro-inflammatory cytokine synthesis. The MAPK Assay was performed to evaluate the effect of As_2_O_3_ on the total expression of MAPK (Erk1/2). The Muse^®^ Cell Analyser was used to measure the amount of MAPK activated in MCF-7 cells treated for 24 h with 11 µM and 32 µM As_2_O_3_, 100 µM CoCl_2_ and 100 µM curcumin. The results in Figure 9 demonstrated higher levels of activated MAPK in the untreated MCF-7 cells (75.35 ± 4.308). The levels significantly decreased to 43.18 ± 3.902 (*p* < 0.01) and 40.23 ± 8.805 (*p* < 0.05) after their treatment with 11 μM and 32 μM of As_2_O_3_, respectively.

### 3.10. As_2_O_3_ Suppresses PI3K Activation in MCF-7 Cells

To investigate the activation and deactivation of the PI3K pathway, the MCF-7 cells were exposed to 11 and 32 µM for 24 h and phospho-PI3K levels were evaluated Muse^®^ Cell Analyzer analyses. As depicted in Figure 10, As_2_O_3_ deactivated the PI3K pathway in a dose-dependent manner. According to Merck-Millipore, “*the Muse^®^ PI3K Activation Dual Detection Kit includes two directly conjugated antibodies, a phospho-specific anti-phospho-Akt (Ser473), Alexa Fluor^®^555 and an anti-Akt, PECy5 conjugated antibody to measure total levels of Akt. This two colour kit is designed to measure the extent of Akt phosphorylation relative to the total Akt expression in any given cell population. By doing such, the levels of both the total and phosphorylated protein can be measured simultaneously in the same cell, resulting in a normalized and accurate measurement of PI3K activation after stimulation*.” The untreated cells exhibited 81.80 ± 2.626 PI3K activation while there was a significant decline of the activated PI3K after treatment with 11 µM reducing it to 53.59 ± 8.073 (*p* < 0.05) and 32 µM to 46.08 ± 3.738 (*p* < 0.01), respectively.

### 3.11. The Upregulation and Downregulation of Survivin Splice Variants during As_2_O_3_-Induced Cell Cycle Arrest and Apoptosis

As shown in Figure 11, this study showed downregulation of the anti-apoptotic survivin variant 3α in the untreated MCF-7 cells. Survivin variant 2B was found to be highly upregulated during As_2_O_3_-induced cell cycle arrest but not cobalt chloride-induced cell cycle arrest, As_2_O_3_-induced and curcumin-induced apoptosis (Figure 11a). During As_2_O_3_ and cobalt chloride-induced G2/M cell cycle arrest and As_2_O_3_-induced apoptosis, survivin 3α was found to be upregulated compared to the untreated cells (Figure 11b). Survivin wild-type was highly expressed in the untreated MCF-7 cells and downregulated in both cell cycle arrest and apoptosis induction (Figure 11c). Our results suggest that survivin variant ΔEx3 is undetectable in untreated and treated MCF-7 cells (Figure 11d). Variant ΔEx3 was only found in colon cancer cells.

### 3.12. The Sub-Cellular Localization of Survivin Protein in Breast Cancer Cell Line MCF-7

Immunohistochemistry was performed to localize and better understand the survivin protein expression in both treated and untreated MCF-7 cells. Survivin protein was localized both in both cytoplasm and nucleus as shown in Figure 12. The anti-survivin antibody detected survivin protein in both untreated (Figure 12d) and treated cells (Figure 12f,h). The highest expression was found in 11 µM As_2_O_3_-treated cells (Figure 12f). This corroborated the RT-PCR results (Figure 11), which showed that survivin 2B was highly expressed during G2/M cell cycle arrest but not apoptosis. There was little to no expression of survivin proteins after treatment with cobalt chloride (J) and curcumin (L).

## 4. Discussion

The purpose of this investigation was to evaluate the potential effect of As_2_O_3_ on the expression of the different survivin splice variants during cell cycle progression and apoptosis of breast cancer cell line, MCF-7. To date, there is a limited number of studies that have directly or indirectly investigated the expression of survivin variants in breast cancer cells. The focus has been on the wild-type only, where its expression has been shown to favour the carcinogenesis process [3]. Additionally, As_2_O_3_ has been used for ages to treat proteolytic leukaemia [15]. In order to understand the expression of survivin splice variant during As_2_O_3_-induced cell cycle arrest and apoptosis, the viability of MCF-7 cells was investigated using MTT Assay and the Muse^®^ Count and Viability Assay. Both confirmed that As_2_O_3_ reduced the viability of the MCF-7 cells in a concentration-dependent manner (Figure 1a and Figure 2f). This is in line with the previous reports [13,16], which showed that As_2_O_3_ suppresses the growth of MCF-7 cells. Kasukabe et al. [16] also showed that Cotylenin A enhanced As_2_O_3_-induced growth inhibition of MCF-7 and MDA-MB-231 cells. The 11 µM As_2_O_3_ suppressed 50% cell growth inhibition while 32 µM of As_2_O_3_ yielded the lowest cell viability against the MCF-7 cells and was then used to investigate apoptosis in this study while 11 µM was used to investigate cell cycle arrest potential of As_2_O_3_ in MCF-7 cells.

As demonstrated in Figure 3a, the untreated cells maintained their epithelial morphological shape. However, the cells treated with As_2_O_3_ exhibited morphological changes typical to apoptosis (Figure 3b–e). As_2_O_3_ reduced the MCF-7 cell numbers in a concentration dependent manner, which can be attributed to characteristics observed from the treated cells. The treated cells displayed increased cellular volumes with the formation of condensed chromatin, raptured nuclear membrane and cytoplasmic blebs. These are characteristics of cells undergoing apoptosis. This data concurred what was demonstrated by Liu et al. (2015) where it was shown that As_2_O_3_ treated cells changed from polygonal to round shape, displaying partly condensed chromatin and the appearance of vacuoles. Fluorescence microscopy further confirmed apoptotic features induced by As_2_O_3_ (Figure 4). When the untreated control cells were stained with DAPI, they showed normal intact nuclei, whereas the treated MCF-7 cells with 11 µM As_2_O_3_ displayed mostly mitotic morphology, suggesting that the cells may be arrested during the G2/M cell cycle arrest checkpoint. As the concentration of As_2_O_3_ increased to 16 µM (Figure 4d). Cell numbers decreased with several apoptotic features such membrane blebbing, DNA fragmentation and nuclear condensation, appearing. The cells that were treated with 32 µM, showed more membrane blebbing and nuclei fragmentation (Figure 4d). This concurred with other studies that reported that As_2_O_3_ induces apoptosis in breast cancer MCF-7 cells [17,18]. As the concentration of As_2_O_3_ increased, the number of apoptotic cells also increased, which confirmed that this compound induces cellular apoptosis in a concentration-dependent manner. As_2_O_3_ has been previously shown to reduce the activity of IKKβ, which is a catalytic subunit of the IKK complex, which was shown to be reduced by As_2_O_3_ [13]. As_2_O_3_ has also been shown to be a Notch-1 inhibitor, inactivating the Notch Signalling pathway in breast cancer [19]. Notch inactivation has been shown to cause down-regulation of antiapoptotic genes such as Bcl-2 and NF-κB, resulting in the inhibition of cell growth, invasion and induction of apoptosis [20].

To further elucidate the cell cycle arrest potential of As_2_O_3_, we examined DNA content using Muse^®^ Cell Cycle Kit. In our study, as it can be seen in Figure 5d, the G2/M percentage of cells after treatment with 11 µM As_2_O_3_ (45.13 ± 0.942) was much higher than the complement of G2/M cells in the control untreated cells (25.73 ± 1.824). Our cell cycle results showed that As_2_O_3_ induced the G2/M cell cycle arrest of MCF-7 cells. Several studies also have reported As_2_O_3_ as a therapeutic agent which can induce cell cycle arrest and expression of certain cell cycle-specific genes in malignant tumours [21,22]. Santarelli et al. [23] reviewed the function of survivin and survivin based treatment strategies and highlighted that survivin is involved in cell cycle regulation, involved in spindle formation. This is in line with our finding that showed that survivin and survivin 2B is upregulated during arsenic trioxide-induced G2M cell cycle arrest, which suggest that this variant be part of the chromosomal passenger complex (CPC). Our study further suggest that the upregulation of survivin 3α may be involved in cellular survival in unfavourable condition. Our study also show that survivin 2B may be an antagonist to both survivin and survivin 3α during As_2_O_3_-induced apoptosis. Additionally, our data showed that As_2_O_3_ induced late apoptosis rather than early apoptosis (Figure 6d). The same trend was previously observed in the breast cancer cell line MCF-7 [24] and other cell lines which include T80, HEY and SKOV3 ovarian cancer cells [25]. Our data suggest that As_2_O_3_ induced the extrinsic pathway as As_2_O_3_ was shown to have no effect on the mitochondrial-mediated apoptotic pathway (Figure 7d). This is in line with another study that showed the same results where HepG2 cells were exposed to As_2_O_3_ for a period of 7 days [26]. The caspases (cysteine-directed aspartate-specific proteases) are important in directing the process of programmed cell death in response to apoptotic signals [27]. In our study, Multi-Caspase Assay revealed that 32 µM As_2_O_3_ lead to activation of several caspases including (1, 3, 4, 5, 6, 7, 8 and 9) and this shows that As_2_O_3_ induces caspase-dependent apoptosis because all the activated caspases include the initiator caspases (caspase 8 and 9) and execution caspases (3 and 7) (Figure 8d).

To further characterize the mechanism by which As_2_O_3_ suppress cell growth of MCF-7 cells, intracellular pathways known to be activated during proliferation of cancer cells were analysed. Many of these signal transduction pathways involve activation of protein serine/threonine kinase of extracellular signal-regulated kinases (ERK) family by oestrogen and involved in cellular growth [28]. To determine the effect of As_2_O_3_ on these cell survival pathways, the MCF-7 cells were exposed to 11and 32 µM As_2_O_3_ for 24 h and the levels of MAPK and PI3K activation were determine using the Muse^®^ Cell Analyser. The MAPK (Figure 9f) and PI3K (Figure 10f) were significantly deactivated during As_2_O_3_-induced cell cycle arrest and apoptosis. Previously, As_2_O_3_ was shown to inhibit the activation of MAPK and PI3K pathways in U118-MG cells and gastric cancer cells [29,30]. As_2_O_3_ has also been demonstrated to inhibit lung adenocarcinoma cell line (H1355) growth by down-regulating survivin expression through the activation of p38 and c-Jun N-terminal kinases (JNK) pathways [14]. Inhibition of PI3K or ERK signalling led to explicit inhibition of survivin expression. However, pre-treatment with p38 MAPK inhibitor led to up-regulated survivin levels. PI3K, ERK and p38 MAPK are all involved in regulating survivin expression and up-regulation of survivin may provide a potential cancer therapy [31]. For the first time, this study has demonstrated the involvement of the novel survivin splice, especially, survivin 2B in the inactivation of the two cell survival pathways that are involved in breast cancer development.

Reverse-Transcription Polymerase Chain Reaction (RT-PCR) was used to analyse the differential expression of survivin spliced variants during As_2_O_3_-induced cell cycle arrest and apoptosis in the breast MCF-7 cancer cells. Although expression of survivin during cell cycle progression has been studied, little is known about the expression of spliced variants, especially in As_2_O_3_-mediated cell cycle arrest. Survivin 2B is a variant that is pro-apoptotic or anti-apoptotic depending on the type and stage of cancer. In this study, the expression of variant 2B was undetectable in the untreated MCF-7 cells but upregulated during As_2_O_3_-induced cell cycle arrest (Figure 11a) and the expression was downregulated during As_2_O_3_-induced apoptosis. This data suggests that survivin 2B has a role in cell cycle regulation than apoptosis. Previously, a more variable expression of survivin 2B level was found at different breast cancer stages [9]. Survivin wild-type was found to be highly expressed in the MCF-7 cells and the expression was decreased during As_2_O_3_-induced G2/M cell cycle arrest and apoptosis. Our data revealed low expression of the anti-apoptotic survivin variant 3α in MCF-7 cells (Figure 11b). Its expression was found to be upregulated during As_2_O_3_-induced cell cycle arrest and apoptosis.

The findings of this study agree with the previous studies that have shown that apoptosis induction decreases the expression of survivin wild-type [32,33]. This study showed that survivin ΔEx3 is undetectable in MCF-7 cells and during As_2_O_3_-induced cell cycle arrest and apoptosis but was detected in caco2 colon cancer cells. This supports the previous report that showed little or no expression of ΔEx3 in breast carcinoma [34]. Vegran et al. [35], revealed that survivin wild-type, survivin-ΔEx3 and survivin 2B have no prognostic role in breast carcinoma. There is concrete evidence showing that survivin variants are highly expressed in cancer cells including MCF-7 cells and demonstrated a cell-cycle dependent expression with a noticeable increase in the G2/M phase [36]. Our study, for the first time suggests that survivin splice variants are differentially expressed in MCF-7 during both As_2_O_3_-induced cell cycle arrest and apoptosis. These results may suggest that the mode of action of As_2_O_3_ in the inhibition of cell proliferation may be dependent on the express of different survivin splice variants. Furthermore, the role of survivin in breast cancer progression may be boundless to its role in the inhibition of apoptosis, which is a hallmark of cancer. In this study, we have documented for the first time that Survivin 2B is involved in As_2_O_3_-mediated G2M cell cycle arrest in MCF-7 cells. Previously, G2M cell cycle arrest has been shown to involve the inactivation of PI3K/Akt and ERα pathways [37]. Arsenic trioxide was found to upregulate survivin 2B during G2M cell cycle arrest which was associated with the deactivation of PI3K/Akt pathway. This suggests that strategies targeting survivin 2B and PI3K/Akt may be beneficial for anticancer drug development against breast cancer. Li et al. (2009) showed that As2O3 induces apoptosis and G2M cell cycle arrest by inhibiting PI3K/Akt signalling by Cbl and p53 activation. Our study suggests that survivin 2B plays a role in As_2_O_3_-mediated PI3K/Akt inactivation and this variant can be targeted for anticancer drug development against breast cancer.

Human survivin is located on chromosome 17q25, approximately 3% of the distance from the telomere and it comprises three introns and four exons, encoding 142 amino acids, including one copy of the BIR, essential for apoptosis inhibition and 16.5 kDa molecular weight [38]. In our study, we performed an immunocytochemical analysis of the expression of survivin proteins in breast cancer MCF-7 cells. Survivin proteins were expressed in the nucleus and cytoplasm of MCF-7 cells. The expression significantly increased during As_2_O_3_-induced G2/M cell cycle arrest (Figure 12f) and low expression was observed in As_2_O_3_-induced apoptosis (Figure 12h). These data is consistent with the results from conventional PCR analysis of survivin 2B mRNA expression in MCF-7 cells (Figure 11a). This known but survivin 2B may have significant role in these processes than initially thought. Survivin wild-type is multifunctional, it has been implicated in cell cycle arrest as well as in the inhibition of apoptosis. In this study only arsenic trioxide-induced G2M cell cycle arrest upregulated wild-type survivin while arsenic trioxide-induced-apoptosis downregulated wild-type survivin. Survivin wild-type upregulation during G2M cell cycle arrest has been reported to favour cancer cell survival [39] and this may be the case with survivin 3α.

## 5. Conclusions

As_2_O_3_ inhibited the growth and affected the morphology of the MCF-7 breast cancer cells in a concentration-dependent manner and this further confirms that this compound has anti-tumour activities against MCF-7 breast cancer cells through the inhibition of growth and cell proliferation. The Muse Cell Analyser showed that As_2_O_3_-induced G2/M cell cycle arrest and promoted caspase-dependent apoptosis of MCF-7 cells without causing any damage to the mitochondrial membrane. Deactivation of MAPK and PI3K was accompanied by upregulation of survivin 2B and 3α variants in breast cancer cell line, MCF-7. Survivin wildtype has been reported as a therapeutic target, involvement of survivin 2B in anticancer activities and inhibition of cell survival pathways is a promising finding for future drug development.

## Figures and Tables

**Figure 1 genes-10-00041-f001:**
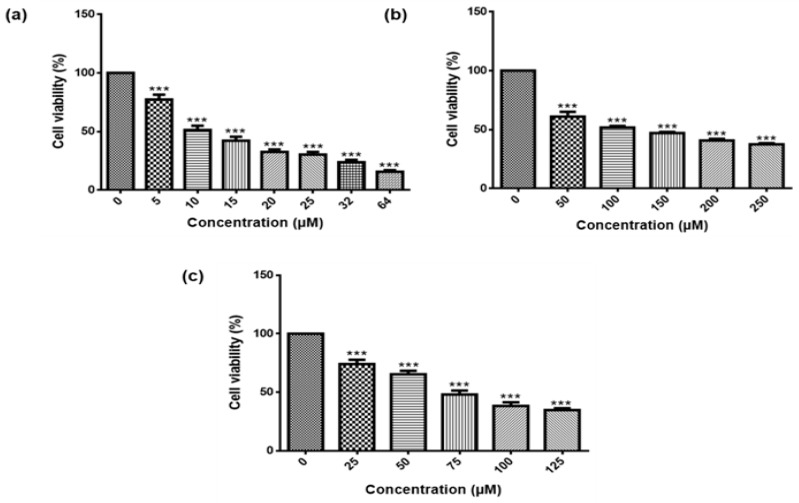
The effect of different concentrations of As_2_O_3_ (**a**), cobalt chloride (**b**) and curcumin (**c**) on the viability of the MCF-7 breast cancer cells. Cell viability was determined with the MTT assay. Shown are representative data of three independent experiments (percentage mean ± SEM). The differences were found to be statistically significant (*** *p* < 0.001).

**Figure 2 genes-10-00041-f002:**
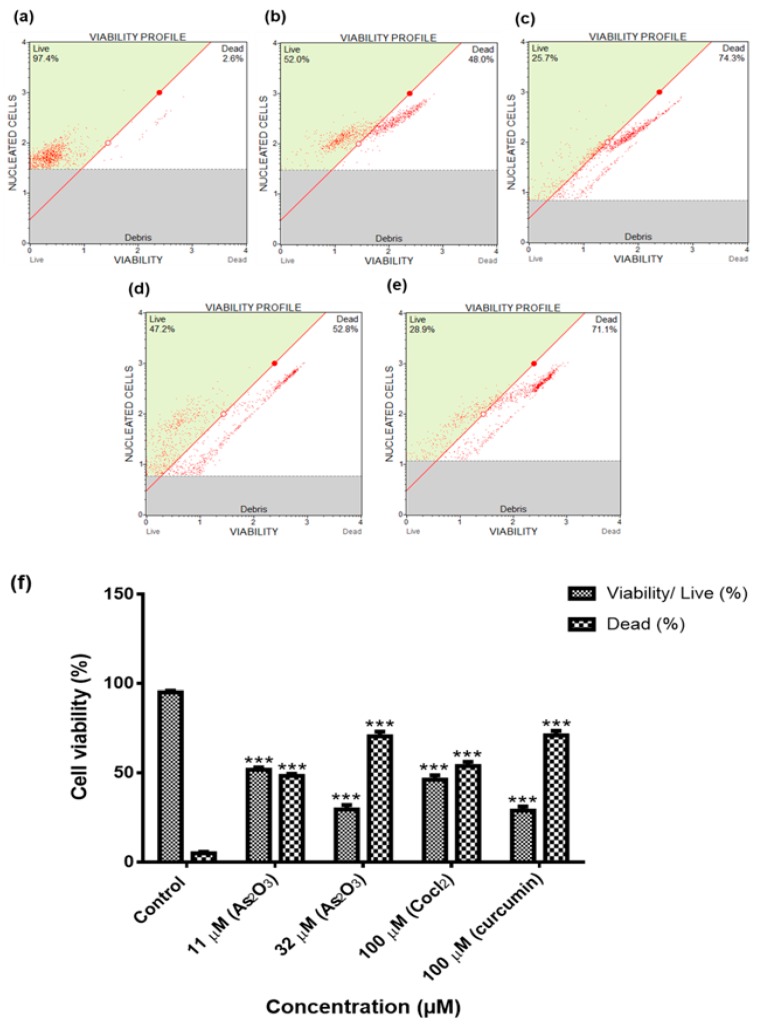
The confirmation of the IC_50_s and apoptosis inducing concentrations using the MUSE^®^ Count and Viability Assay (**a**–**e**). Shown are representative data of three independent experiments [percentage mean ± SEM] (**f**). The difference were found to be statistically significant (*** *p* < 0.001).

**Figure 3 genes-10-00041-f003:**
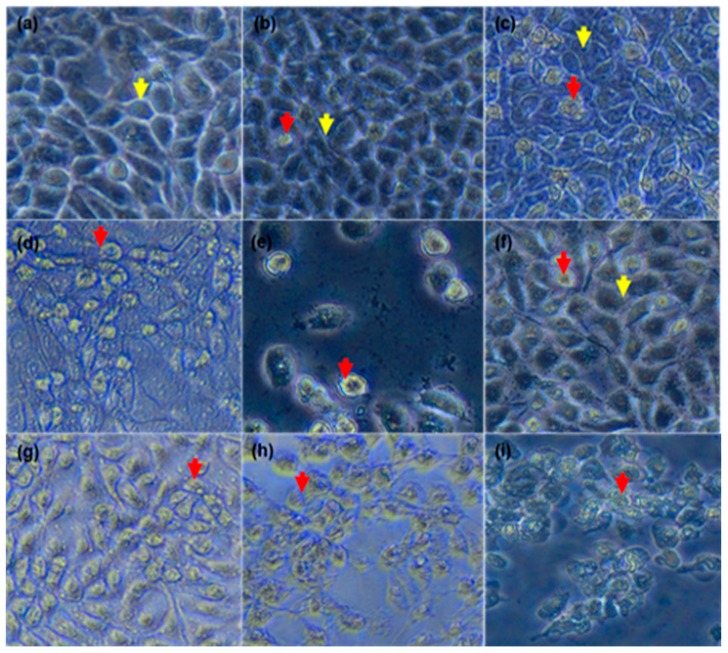
The effect of As_2_O_3_ on the MCF-7 cell morphology. The normal light microscopy images of MCF-7 untreated cells (**a**), treated cells with concentrations of As_2_O_3_ [(**b**)—4 µM, (**c**)—11 µM, (**d**)—16 µM and (**e**)—32 µM], cobalt chloride [(**f**)—50 µM and (**g**)—100 µM] and curcumin [(**h**)—50 µM and (**i**)—100 µM]. The yellow arrows indicate the normal shape of MCF-7 cell and the red arrows show the apoptotic cells.

**Figure 4 genes-10-00041-f004:**
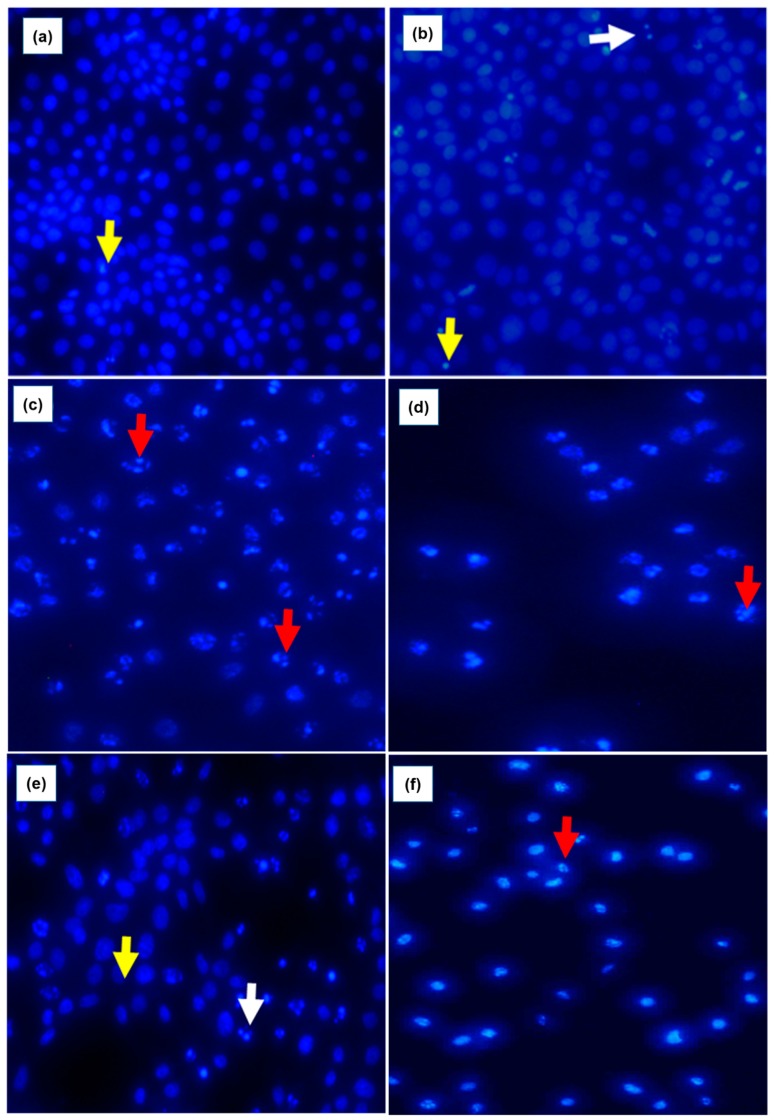
Nuclear morphology of MCF-7 cells after DAPI staining. The fluorescence microscopy images of the untreated MCF-7 cells (**a**). MCF-7 cells treated with various concentrations of As_2_O_3_ [(**b**)–4 µM, (**c**)—16 µM, (**d**)—32 µM], cobalt chloride [(**e**)—100 µM] and curcumin [(**f**)—100 µM] for 24 h. The samples were analysed using an Eclipse Ti-U fluorescence microscope (Nickon Instruments Inc.) and images were captured at 20 × magnification. The white arrows indicate intact nucleus, yellow arrows indicate the mitotic cells and the red arrows show the apoptotic cells.

**Figure 5 genes-10-00041-f005:**
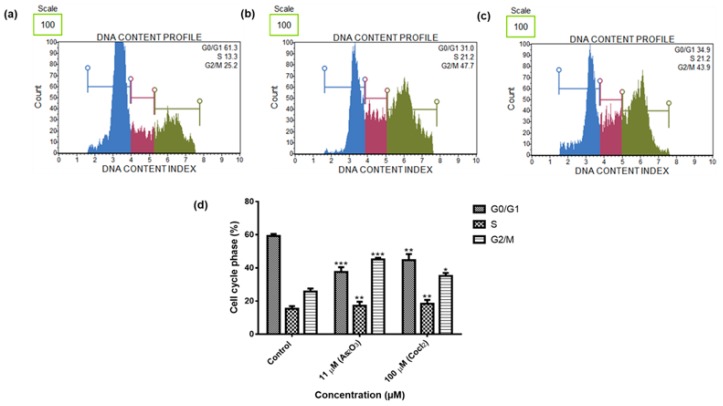
Cell cycle profiles of the MCF-7 cells after 24 h treatment: (**a**)—Control, (**b**)—11 μM As_2_O_3_, (**c**)—100 μM cobalt chloride. Shown are representative data of three independent experiments [percentage mean ± SEM] (**d**). The difference were found to be statistically significant (* *p* < 0.05, ** *p* < 0.01 and *** *p* < 0.001).

**Figure 6 genes-10-00041-f006:**
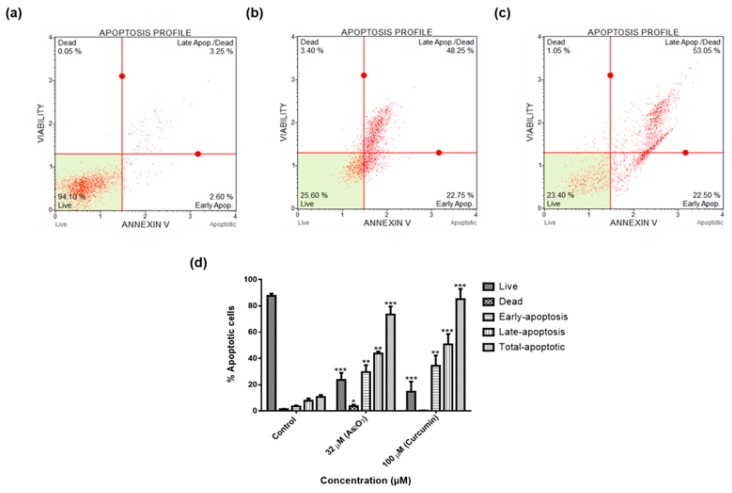
Apoptosis profiles of MCF-7 cells after 24 h treatment: (**a**)—Control, (**b**)—32 µM As_2_O_3_, (**c**)—100 µM curcumin. Shown are representative data of three independent experiments [percentage mean ± SEM] (**d**). The difference were found to be statistically significant (* *p* < 0.05, ** *p* < 0.01 and *** *p* < 0.001).

**Figure 7 genes-10-00041-f007:**
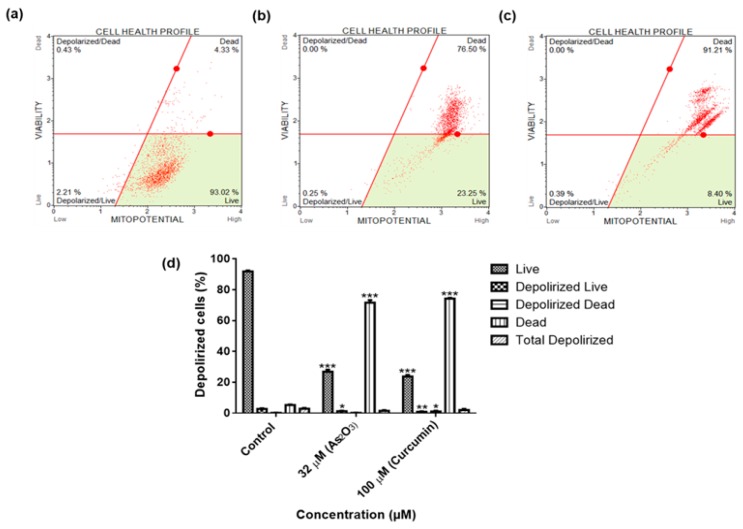
Mitochondrial membrane integrity profiles of the MCF-7 cells after 24 h treatment: (**a**)—Control, (**b**)—32 µM As_2_O_3_, (**c**)—100 µM curcumin. Shown are representative data of three independent experiments [percentage mean ± SEM] (**d**). The difference were found to be statistically significant (* *p* < 0.05, ** *p* < 0.01 and *** *p* < 0.001).

**Figure 8 genes-10-00041-f008:**
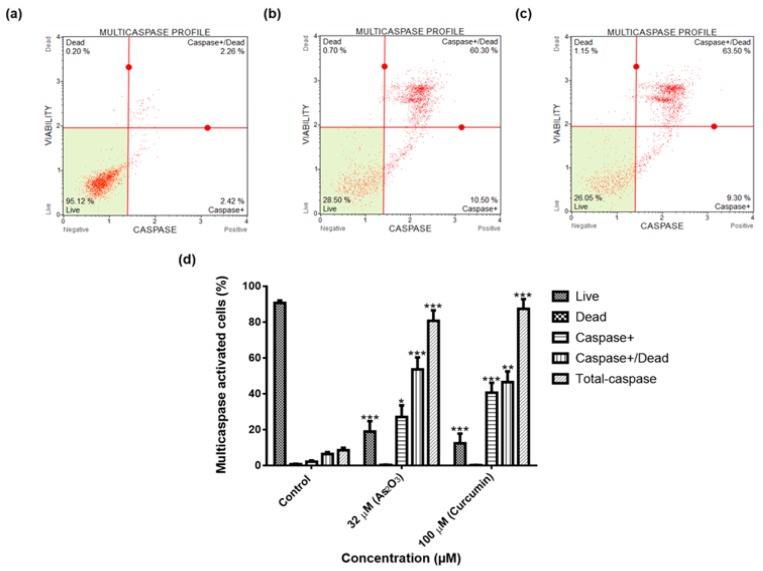
Multi-Caspase profiles of the MCF-7 cells after 24 h treatment: (**a**)—Control, (**b**)—32 µM arsenic trioxide and (**c**)—100 µM curcumin. Shown are representative data of three independent experiments (**d**) [percentage mean ± SEM]. The difference were found to be statistically significant (* *p* < 0.05, ** *p* < 0.01 and *** *p* < 0.001).

**Figure 9 genes-10-00041-f009:**
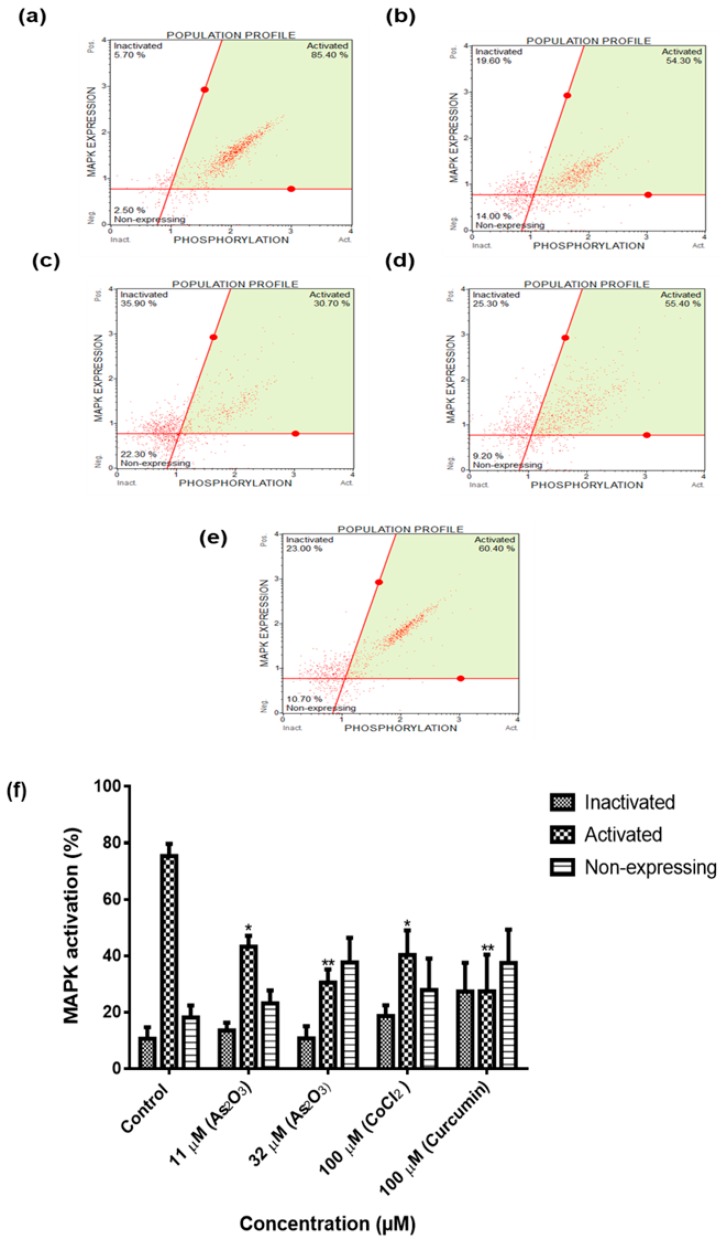
MAPK population profiles of the MCF-7 cells after 24 h treatment: (**a**)—Control, (**b**)—11 µM As_2_O_3_, (**c**)—32 µM As_2_O_3_, (**d**)—100 μM cobalt chloride, 100 μM curcumin. Shown are representative data of three independent experiments [percentage mean ± SEM] (**f**). The difference were found to be statistically significant (* *p* < 0.05, ** *p* < 0.01 and *** *p* < 0.001).

**Figure 10 genes-10-00041-f010:**
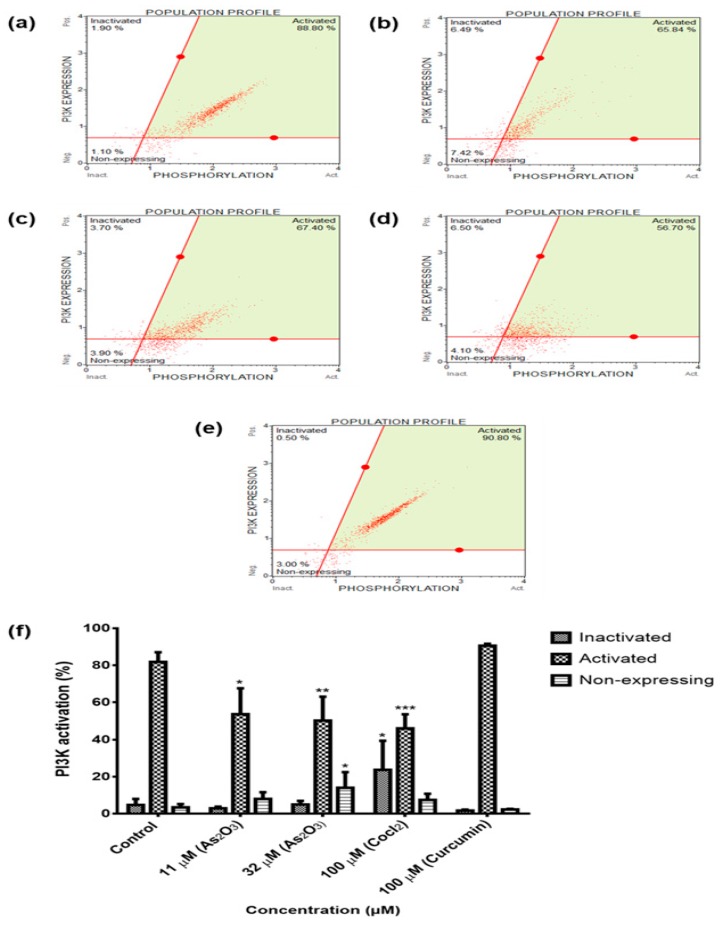
PI3K population profiles of the MCF-7 cells after 24 h treatment: (**a**)—Control, (**b**)—11 µM As_2_O_3_, (**c**)—32 µM As_2_O_3_, (**d**)—100 µM cobalt chloride and (**e**)—100 µM curcumin. Shown are representative data of three independent experiments [percentage mean ± SEM] (**f**). The difference were found to be statistically significant (* *p* < 0.05, ** *p* < 0.01 and *** *p* < 0.001).

**Figure 11 genes-10-00041-f011:**
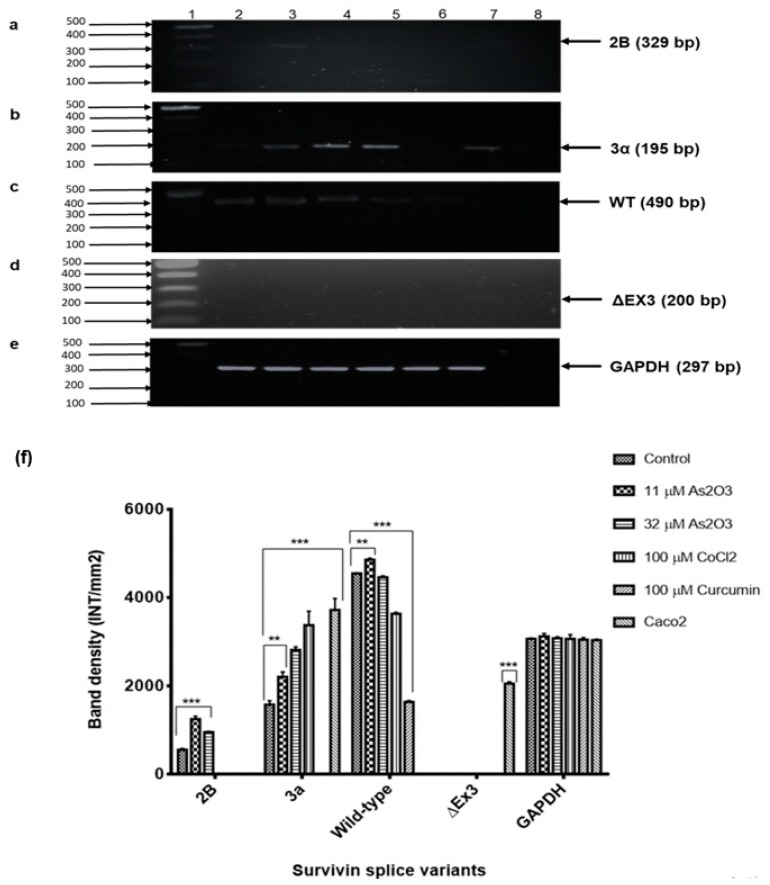
RT-PCR analyses of survivin splice variants expression, (**a**)—2B, (**b**)—3α, (**c**)—wild-type and (**d**)—ΔEx3 in MCF-7 breast cancer cells after treating with the As_2_O_3_, cobalt chloride and curcumin for 24 h. GAPDH (**e**) was used as the loading control. The images were captured using Chemidoc XRS image analyser (BioRad, USA). The band density differences of survivin splice variants (**f**). Shown are representative data of three independent experiments (percentage mean ± SEM). The difference were found to be statistically significant (* *p* < 0.05, ** *p* < 0.01 and *** *p* < 0.001). The density was measured using Quanty-One software (BioRad, USA).

**Figure 12 genes-10-00041-f012:**
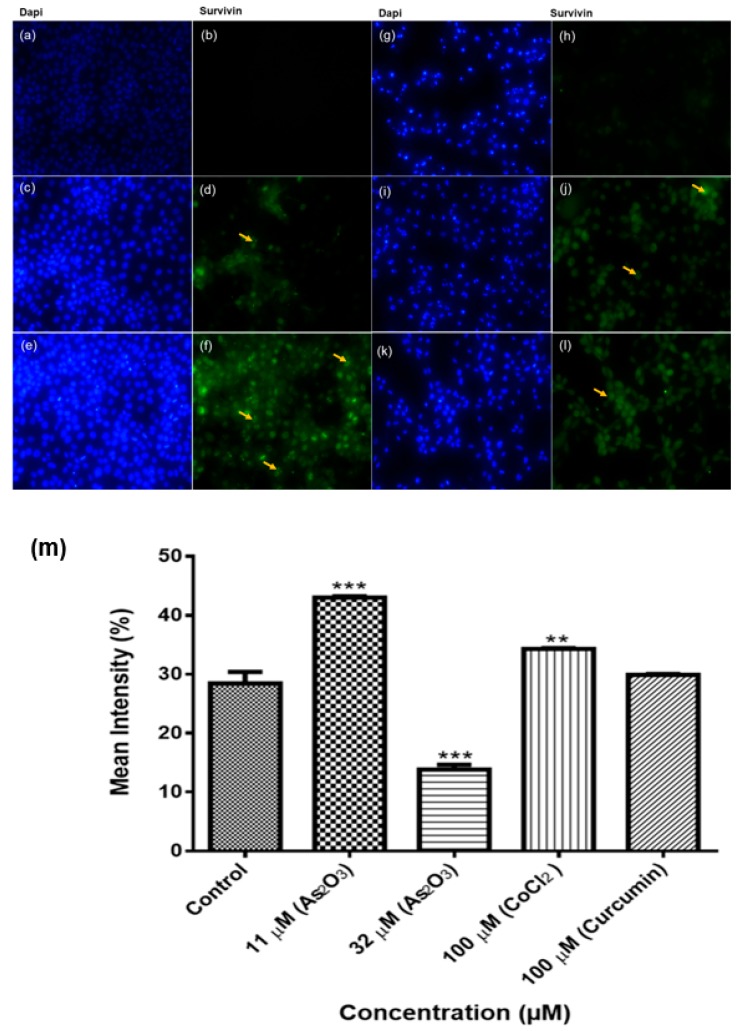
Immunostaining of survivin protein in MCF-7cells. The unstained untreated cells (**a**,**b**) and stained untreated cells (**c**,**d**). The cells were treated with 11 (**e**,**f**) and 32 (**g**,**h**) µM As_2_O_3_, 100 µM cobalt chloride (**i**,**j**) and 100 µM curcumin (**k**,**l**) for 24 h. The cells were counterstained with DAPI. The samples were analysed using an Eclipse Ti-U fluorescence microscope (Nickon Instruments Inc.) and images were captured at 20 × magnification. The yellow arrows show the survivin protein. The mean intensity percentage of the survivin proteins expression (**m**). Shown are representative data of three independent experiments (percentage mean ± SEM). The difference were found to be statistically significant (* *p* < 0.05, ** *p* < 0.01 and *** *p* < 0.001). The mean intensity was measure using the Image J software (https://imagej.nih.gov/ij/docs/index.html).

**Table 1 genes-10-00041-t001:** Different survivin splice variants and their primers.

Variant	Forward Primer	Reverse Primer
Survivin wild-type	5′ TAAGAGGGCGTGCGCTCCC 3′	5′-ATGGCACGGCGCTCTTTCTC-3′
Survivin 3α	5′ CAGGGAGGGACTGGAAGCA 3′	5′ CTCCTGAAACTCCTGGAGGA 3′
Survivin 2B	5′ TGGGAGCCAGATGACGACCC 3′	5′ CCCTGGAAGTGGTGCAGCCA 3′
Survivin ∆EX3	5′ CTGGGAGCCAGATGACGACC 3′	5′ AAGGCTGGTGGCACCAGGGA 3′
GAPDH	5′ AGCTGAACGGGAAGCTCACT 3′	5′ AGCTGAACGGGAAGCTCACT 3′

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
