# Peer review of "Survivin Splice Variants in Arsenic Trioxide (As2O3)-Induced Deactivation of PI3K and MAPK Cell Signalling Pathways in MCF-7 Cells"

_genes, 2019, doi:10.3390/genes10010041_

Round 1
Reviewer 1 Report
I have now completed reviewing the research article, and the figures thoroughly require clarity enhancement. Some of the figure legends are not even visible for understanding.
What was the reason for inclusion of 50 and 100 microM of cir cumin and cobalt chloride in the result section 3.4?
And why wasn't cobalt chloride not included in all the assays? why wasn't it in-consistant?
Figure 4 requires higher magnification. The features such as chromatin condensation and nucleus fragmentation is not visible.
There is a control missing in the first study i.e. cell viability (how come there is no group without treatments?)
In the results section, 3.11 the authors have mentioned that Survivin 3 alpha was unregulated in AS2O3 induced apoptosis. How can an anti apoptotic protein be unregulated during treatment with a pro-apoptotic agent?
In fact even the wild variant of Survivin was up-regulated in AS2O3 treatment. How can any anti apoptotic protein be up-regulated with an anti cancer drug?
The same goes even with figure 12. Also the figure 12 is highly confusing to refer.
The authors have just stated in the discussion like 465 that: the findings of this study agree with the previous studies that have shown that apoptosis induction decreases the expression of survivin wild-type.
But the results indicate that the expression of Survivin was increased in the pro apoptotic group. How is this possible?
The study is completely confusing and misleading.
Author Response
Comment: What was the reason for inclusion of 50 and 100 µM of curcumin and cobalt chloride in the result section 3.4, respectively?
Response: Both curcumin and cobalt chloride were used as positive controls for apoptosis and cell cycle arrest analysis, respectively. In the cell viability results, 100µM cobalt chloride reduced the viability of MCF-7 cells by 50% and therefore, this was a concentration we used in the cell cycle assays. Similarly, using curcumin, the viability of MCF-7 cells was reduced by 50% between 75µM and 100µM curcumin and the latter was used for all the subsequent apoptosis related assays. The figure has been revised to include only the 100µM concentrations. The figure has been made bigger to make the apoptotic features visible.
Comment: And why wasn't cobalt chloride not included in all the assays? Why wasn't it in-consistant?
Response: Cobalt chloride was only used as positive control for cell cycle arrest assays as it is well known to induce the G2/M cell cycle arrest while curcumin was used as a positive control for apoptosis assays. These two positive controls were therefore used for the relevant experiments.
Comment: Figure 4 requires higher magnification. The features such as chromatin condensation and nucleus fragmentation is not visible.
Response: Due to infrastructure constraints, we could only manage 20X images but as the arrows show in the figures, we believe that these images do demonstrate nuclear condensation and fragmentation when compared to the untreated control cells, which showed intact and bigger nuclei. We also demonstrated the induction of apoptosis using flow cytometry.
Comment: There is a control missing in the first study i.e. cell viability (how come there is no group without treatments?)
Response: As demonstrated in all the figures, including the cell viability, microscopy, flow cytometry-Annexin V and survival pathways, the untreated control cells were included. In figure 1, the 0µM cells were the control cells, which were grown in medium only.
Comment: In the results section, 3.11 the authors have mentioned that Survivin 3 alpha was unregulated in As2O3 induced apoptosis. How can an anti apoptotic protein be unregulated during treatment with a pro-apoptotic agent?
Response: In this study, we confirmed that survivin 3 alpha is upregulated during arsenic trioxide-induced apoptosis. There has been little and inconsistent data on the function of the survivin splice variants but we suspect that this variant may be involved in apoptosis resistance as there is little information on its function. This opens more debate on the function of the survivin 3α in breast cancer cells.
Comment: In fact even the wild variant of Survivin was up-regulated in AS2O3 treatment. How can any anti apoptotic protein be up-regulated with an anticancer drug?
Response: Survivin wild-type is multifunctional, it has been implicated in cell cycle arrest as well as in the inhibition of apoptosis. In this study only arsenic trioxide-induced G2M cell cycle arrest upregulated wild-type survivin while arsenic trioxide-induced-apoptosis downregulated wild-type survivin. Survivin wild-type upregulation during G2M cell cycle arrest has been reported to favour cancer cell survival (Jandial et al., 2017) and this may be the case with survivin 3 alpha.
Jandial et al. (2017) Induction of G2M Arrest by Flavokawain A, a Kava Chalcone, Increases the Responsiveness of HER2-Overexpressing Breast Cancer Cells to Herceptin. Molecules 22 (3): 462.
Comment: The same goes even with figure 12. Also the figure 12 is highly confusing to refer.
Response: Figure 12 shows that treatment with the 32µM As2O3 inducing concentration upregulated survivin and we suggest that this is mostly attributed to upregulation of survivin 2B, probably to a lesser extent, wild-type survivin.
Comment: The authors have just stated in the discussion like 465 that: the findings of this study agree with the previous studies that have shown that apoptosis induction decreases the expression of survivin wild-type. But the results indicate that the expression of Survivin was increased in the pro apoptotic group. How is this possible? The study is completely confusing and misleading.
Response: A few citations have been added to support our claim that induction of G2M cell cycle arrest and apoptosis upregulates and downregulates survivin wild-type expression. For the first time, this study demonstrated that the functions of the survivin splice variants are not fully understood. The study clearly showed that arsenic trioxide can either induce G2M cell cycle arrest or apoptosis that is associated with regulation of different survivin splice variants in MCF-7 and deactivation of cell survival pathways. Survivin 2B may be crucial for the arsenic trioxide G2M cell cycle arrest while 3α may be involved in cell survival mechanisms in response to arsenic trioxide.
Reviewer 2 Report
Row 36-37. Cancer is a worldwide problem and I believe this is common knowledge. I think is best to remove this section.
Row 39-41. The resistance to apoptosis can be caused by many other mechanisms. Please expand this section a little more.
Row 59-61. This contains important mistakes. “Targeting survivin has yielded potential therapeutic drugs such as cisplatin (Kumar et al., 2012), erlotinib (Okamoto et al., 2012), doxorubicin (Feversani et al., 2014) and rapamycin (Koike et al., 2014).” First of all, they are already drugs, and not potential therapeutic drugs. Secondly, cisplatin was serendipitously discovered while studying the effect of electric current in E. Coli. Doxorubicin and rapamycin are antibiotics isolated from streptomyces in the 1960s. None of them were design to target survin!!! Please remove this section.
Some previous articles already studied the effect of arsenic trioxide on apoptosis in MCF-7 cells. Some of them are discussed by the authors (see Arsenic trioxide induces the apoptosis of human breast cancer MCF-7 cells through activation of caspase-3 and inhibition of HERG channels), but the authors should highlight the differences and the originality of their own paper.
Also, some other paper on the same subject should be presented and discussed:
Arsenic Trioxide Inhibits Cell Growth and Induces Apoptosis through Inactivation of Notch Signaling Pathway in Breast Cancer
Cotylenin A and arsenic trioxide cooperatively suppress cell proliferation and cell invasion activity in human breast cancer cells
Arsenic Trioxide-Induced Growth Arrest of Breast Cancer MCF-7 Cells Involving FOXO3a and IκB Kinase β Expression and Localization
How was IC50 calculated? Please add some method description. Correct the presenting style “The IC50 for As2O3 found to be between 10 µM (51.22±3.064) and 15 µM (42.10± 3.747)” This is confusing. It should be IC50 is X±y
I don’t understand the choice of Cobalt chloride as control. Please provide some justification!
Row 177. The authors declare the measurement of pAkt and Akt, but the results are missing. The authors declare that As2O3 deactivated PI3K pathways, but as long as the effect on pAkt was not measure this is just a hypothesis. The activity of PTEN is also an important factor that has a great impact on Akt activation.
The discussion section has to be modified. The authors need to prove the originality of their findings, considering that many effect of As2O3 on MCF-7 are already known. Also, the authors should discuss more on PI3K and MAPK data and correlation with surviving.
In the conclusion section “Further studies concerning the controlling mechanisms of surviving splice variants expression and function in normal and cancerous cells will help to clarify survivin’s biology and consequently to develop innovative strategies for selective survivin inhibition”. I suggest a more objective style. The article has a little impact on surviving mechanisms and the results show no insight on finding selective survivin inhibition.
The style of the article is not always the same. The manuscripts needs a careful check. For example, Fig 10 caption. a-Control and [f], while in Fig 11 is (a) or (b). The authors use As2O3, as well as arsenic trioxide. Use a unitary style!
Author Response
Comment: Row 36-37. Cancer is a worldwide problem and I believe this is common knowledge. I think is best to remove this section.
Response: The section was removed as suggested.
Comment: Row 39-41. The resistance to apoptosis can be caused by many other mechanisms. Please expand this section a little more.
Response: An expansion has been added in the introduction section.
Comment: Row 59-61. This contains important mistakes. “Targeting survivin has yielded potential therapeutic drugs such as cisplatin (Kumar et al., 2012), erlotinib (Okamoto et al., 2012), doxorubicin (Feversani et al., 2014) and rapamycin (Koike et al., 2014).” First of all, they are already drugs, and not potential therapeutic drugs. Secondly, cisplatin was serendipitously discovered while studying the effect of electric current in E. coli. Doxorubicin and rapamycin are antibiotics isolated from streptomyces in the 1960s. None of them were design to target survivin!!! Please remove this section.
Response: This was an oversight, this section was removed and the article was read again.
Comment: Some previous articles already studied the effect of arsenic trioxide on apoptosis in MCF-7 cells. Some of them are discussed by the authors (see Arsenic trioxide induces the apoptosis of human breast cancer MCF-7 cells through activation of caspase-3 and inhibition of HERG channels), but the authors should highlight the differences and the originality of their own paper.
Response: In the article, we have shown that arsenic trioxide (As2O3) regulates survivin variants during cell cycle arrest and apoptosis in MCF-7 cells. Survivin 2B may be required during the G2/M cell cycle arrest but not involved in As2O3-induced apoptosis. Survivin 3α was found to be involved in both As2O3-induced G2/M cell cycle arrest and As2O3-induced apoptosis. Lastly, survivin 2B and survivin 3α may be implicated in As2O3-induced deactivation of MAPK and PI3K in breast cancer cells MCF-7.
Comment: Also, some other paper on the same subject should be presented and discussed:
Arsenic Trioxide Inhibits Cell Growth and Induces Apoptosis through Inactivation of Notch Signaling Pathway in Breast Cancer.
Cotylenin A and arsenic trioxide cooperatively suppress cell proliferation and cell invasion activity in human breast cancer cells.
Arsenic Trioxide-Induced Growth Arrest of Breast Cancer MCF-7 Cells Involving FOXO3a and IκB Kinase β Expression and Localization.
Response: The suggested articles have been read and used in the discussion of our results.
Comment: How was IC50 calculated? Please add some method description. Correct the presenting style The IC50 for As2O3 found to be between 10 µM (51.22±3.064) and 15 µM (42.10± 3.747)” This is confusing. It should be IC50 is X±y.
Response: This has been addressed under section 2.3. The IC50 was found to be 11µM (51.78 ± 1.243).
Comment: I don’t understand the choice of Cobalt chloride as control. Please provide some justification.
Response: Under section 2.3, the choices of using the positive controls were justified. Cobalt chloride was used as a positive control for cell cycle assays only as it is known to induce the G2/M cell cycle arrest while curcumin was used as a positive control for apoptosis. These two positive controls were therefore used for the relevant experiments.
Comment: Row 177. The authors declare the measurement of pAkt and Akt, but the results are missing. The authors declare that As2O3 deactivated PI3K pathways, but as long as the effect on pAkt was not measure this is just a hypothesis. The activity of PTEN is also an important factor that has a great impact on Akt activation.
Response: The Muse® PI3K Activation Dual Detection Kit includes two directly conjugated antibodies, a phospho-specific anti-phospho-Akt (Ser473), Alexa Fluor®555 and an anti-Akt, PECy5 conjugated antibody to measure total levels of Akt. This two color kit is designed to measure the extent of Akt phosphorylation relative to the total Akt expression in any given cell population. The kit was used to measure the levels of phopho-Akt, therefore we were able to detect the effect of As2O3 on PI3 activation or deactivation.
Comment: The discussion section has to be modified. The authors need to prove the originality of their findings, considering that many effect of As2O3 on MCF-7 are already known. Also, the authors should discuss more on PI3K and MAPK data and correlation with surviving.
Response: This has been done in the discussion section.
Comment: In the conclusion section “Further studies concerning the controlling mechanisms of surviving splice variants expression and function in normal and cancerous cells will help to clarify survivin’s biology and consequently to develop innovative strategies for selective survivin inhibition”. I suggest a more objective style. The article has a little impact on surviving mechanisms and the results show no insight on finding selective survivin inhibition.
Response:
Comment: The style of the article is not always the same. The manuscripts needs a careful check. For example, Fig 10 caption. a-Control and [f], while in Fig 11 is (a) or (b). The authors use As2O3, as well as arsenic trioxide. Use a unitary style!
Response: The inconsistences have been addressed.
Reviewer 3 Report
This is a remarkable study investigating the effects of the use of Arsenic Trioxide in Survivin expression, a well-known anti-apoptotic protein with several biological functions, in breast cancer MCF-7 cells.
The techniques utilized were appropriate and described with plenty details. This is a well-designed study with rigorous methods. The discussion is well-balanced, and the statements are supported by the data.
The work is interesting because, as the Authors stated, provide the first evidence showing that Survivin 2B splice variant is involved in the regulation of Arsenic Trioxide-dependent cell cycle arrest.
I suggest only to add some considerations regarding the potential of Survivin as a therapeutic target for new anticancer treatments, as described in a recent review [1].
[1]: Santarelli A. et al. Survivin-Based Treatment Strategies for Squamous Cell Carcinoma. Int J Mol Sci. 2018 Mar 24;19(4). pii: E971. doi: 10.3390/ijms19040971.
Author Response
Comment: I suggest only to add some considerations regarding the potential of Survivin as a therapeutic target for new anticancer treatments, as described in a recent review [1].
[1] Santarelli A. et al. Survivin-Based Treatment Strategies for Squamous Cell Carcinoma. Int J Mol Sci. 2018 Mar 24;19 (4). pii: E971. doi: 10.3390/ijms19040971.
Response: The conclusion has been reworked on to reflect possible future drug developments.
Round 2
Reviewer 1 Report
as the authors have made changes which are satisfactory, the artcle cAn now be accepted
Reviewer 2 Report
The authors made most of the suggested changes and improved their manuscript. Still there are some editing error to be corrected.